# Bolstering the Measurement of Racial Inequity of COVID-19 Vaccine Uptake

**DOI:** 10.3390/vaccines11040876

**Published:** 2023-04-21

**Authors:** Savanah Russ, John Bramley, Yu Liu, Irena Boyce

**Affiliations:** 1Department of Public Health Sciences, University of Rochester School of Medicine and Dentistry, Rochester, NY 14642, USA; savanah_russ@urmc.rochester.edu (S.R.);; 2UR Medicine Quality Institute, University of Rochester Medical Center, Rochester, NY 14642, USA

**Keywords:** COVID-19 vaccines, race/ethnicity, health equity, racial disparities, public health

## Abstract

Inequities in COVID-19 vaccine uptake by racialized groups have been persistent throughout the vaccine rollout, leading to disparate burdens of COVID-19 outcomes. A cross-sectional study was conducted to determine COVID-19 vaccine uptake across racialized groups within the nine-county Finger Lakes region of New York State in December 2021. Cross-matching and validation were performed across multiple health information systems for the region to reduce the percentage of vaccine records with missing race information. Additionally, imputation techniques were applied to address the remaining missing values. Uptake of ≥1 dose of the COVID-19 vaccine by race was then examined. By December 2021, 828,551 individuals in our study region had received ≥1 dose of the COVID-19 vaccine, with ~25% having missing race values. Cross-matching and validation within existing records reduced this to ~7%. Uptake of ≥1 dose of a COVID-19 vaccine was greatest among individuals identifying as White, followed by those identifying as Black. The application of imputation techniques reduced the percent of missing race values to <1%; however, this reduction did not significantly change the distribution of vaccine uptake across race groups. Utilization of relevant health information systems, accompanied by imputation techniques, stands to greatly reduce the burden of missing race data within vaccine registries, facilitating accurate targeted interventions to mitigate inequities in COVID-19 vaccination.

## 1. Introduction

Disparities across COVID-19 outcomes by racialized groups have been highly prevalent since the start of the pandemic, with higher incidence, hospitalization and mortality rates among racial and ethnic minorities in the United States (U.S.) [1]. In December 2020, two highly effective mRNA COVID-19 vaccines were approved by the U.S. Food & Drug Administration (FDA) and recommended by the Advisory Committee on Immunization Practices (ACIP) for receipt [2,3]. While the introduction of the vaccines has resulted in a decline of severe COVID-19 illness in the U.S., uptake of the vaccine and the subsequent booster vaccines has been suboptimal, with uptake of the primary series to date at 68.8% [4]. Additionally, uptake of these COVID-19 vaccines has not been equal across racialized groups, with the lowest uptake consistently falling among those identifying as Non-Hispanic Black or Non-Hispanic White [5]. However, the full extent of disparate uptake of the COVID-19 vaccine by racialized group is unknown, given the high percentage of missing race and ethnicity information (>25%) within vaccination records across the U.S [6].

Identification of disparate COVID-19 vaccine uptake is crucial for the success of mitigation efforts aimed at limiting excess burden of vaccine-preventable illness. Persistently high levels of missing race within COVID-19 vaccination records can inhibit our ability to understand determinants driving low uptake, and may ultimately result in misguided interventions and failure to address disparate COVID-19 vaccine uptake. The high percentage of missing race information within vaccination records has presented a complex challenge for public health practitioners to address, given that the percent is generally above the accepted threshold of 5%, in which case complete case analysis is accepted [7]. Use of existing health records, as well as robust imputation techniques, serves as a viable option to alleviate these data concerns by using existing information within vaccine records to infer possible race values for those missing. The application of imputation techniques, as compared to complete case analysis, facilitates an increase in the sample size of the data and can lead to more efficient and unbiased estimates for evaluating disparate COVID-19 vaccine uptake. To determine if any unidentified gaps in COVID-19 vaccine uptake exist by race, we aimed to evaluate COVID-19 vaccine uptake while simultaneously reducing the percentage of vaccination records with missing race values within the NYSIIS database for the Finger Lakes region of NY, for a population of over 1.2 million individuals. The goal of these analyses is to enhance the identification of disparities in COVID-19 vaccine uptake by race to inform targeted intervention efforts for increasing uptake and mitigating inequities of severe COVID-19 burden across the region.

## 2. Materials and Methods

Data for this analysis came from the New York State Immunization Information System (NYSIIS). Developed and maintained by the New York State Department of Health following the approval of the Immunization Registry Law by the New York State legislature in 1 January 2008, this state-wide immunization information system mandates the reporting of all immunizations administered to New York State residents < 19 years of age, along with their full immunization history [8]. In March 2020, a state-wide executive order required providers to report COVID-19 immunizations to NYSIIS [9]. Information reported through NYSIIS records includes patients’ race/ethnicity, age, gender, residence, and immunization location.

Our study population consisted of vaccine-eligible residents of the Finger Lakes region of New York (NY) State who received at least one dose of a COVID-19 vaccine between 14 December 2020 and 31 December 2021. The Finger Lakes region of NY consists of nine counties: Genesee, Monroe, Livingston, Ontario, Orleans, Seneca, Wyoming, Wayne and Yates. Residents younger than 5 years of age were excluded, as the vaccine was not recommended for this population until 2022.

To examine uptake of at least one dose of the COVID-19 vaccine by race within each of the respective nine counties, the percentage of uptake was examined within each of the following race groups: White, Black, Asian and Other race. Individuals within the Other race group were those who reported their race as Pacific Islander, American Indian/Alaskan Native or other. Race for each respective record was self-reported to NYSIIS and recorded for each COVID-19 vaccine record. Additional work was performed to decrease the percentage of missing race values within COVID-19 vaccination records in the NYSIIS database for the Finger Lakes region, which included (1) cross-validation of race values from a local health information exchange for the Finger Lakes region (Rochester Health Information Exchange), and (2) cross-matching of race values across individuals with multiple vaccination records in NYSIIS with previously reported race values. This created an enhanced complete case analysis examination of the distribution of vaccine uptake across race groups.

To examine uptake of ≥1 dose of a COVID-19 vaccine by race groups in the region, given it has an even further reduced percentage of missing race values in vaccination records, multiple imputation was performed. For this imputation, both multinomial logistic regression and a machine-learning random forest algorithm were used to confirm the robustness of the values generated. Both imputation methods utilized a probabilistic approach by which probabilities of being White, Black, Asian or Other race were calculated for the remaining records with missing race values [10]. These probabilities were generated through regression-based approaches using an individual’s known ethnicity, age, sex and county of geographic residence as predictors of each respective individual’s probability of being White, Black, Asian or Other race (Figure 1). These probabilities were then summed across each respective race group, along with the number of individuals vaccinated with known race values. To examine if any unidentified gaps in COVID-19 vaccine uptake were prevalent by race, the percent of individuals vaccinated with ≥1 dose by race group after imputation was compared to that of the complete case analysis.

Sensitivity analysis was performed to determine the accuracy of the two imputation techniques. To do so, 7.76% of the COVID-19 vaccine records with known race values were randomly removed. We chose to remove 7.76%, as this was approximately the percentage of vaccination records with missing race values before multiple imputation was utilized within the full, complete case dataset. The two multiple imputation techniques were then applied, and the distribution of COVID-19 vaccine records by race group was compared between the complete dataset and the dataset with newly imputed values.

This work was deemed one of quality improvement by the university’s institutional review board, and thus exempt from the need for human subject approval. All analyses were performed in Rstudio v4.2.1 and Python v3.9.

## 3. Results

In December 2021, 828,551 individuals in the Finger Lakes region of NY State had received ≥1 dose of a COVID-19 vaccine. Of these, 25% had missing race values. After performing cross-validation across both the Rochester Health Information Exchange and existing NYSIIS records, the percentage of individuals with missing race values within their COVID-19 vaccination records was reduced to ~7%. The application of multiple imputation techniques significantly reduced the percentage of individuals with missing race values in NYSIIS COVID-19 vaccination records to <1.0%. However, multiple imputation yielded a similar distribution of vaccine records across race categories as compared to our enhanced complete case analysis (Table 1; Figure 2).

Sensitivity analyses showed the two multiple imputation techniques were highly accurate, with <1% percentage point difference in the percentage of COVID-19 vaccination records within each respective race category between the dataset with imputed values and the enhanced complete case dataset (Table 2).

## 4. Discussion

Among vaccine-eligible individuals residing in the Finger Lakes region in 2021, the majority of those who received ≥1 dose of a COVID-19 vaccine identified as White, followed by those identifying as Black. While the uptake of 83% for those identifying as White represents a higher proportion than White individuals residing in this region (~75%), uptake among Black individuals in the region (8.3%) is lower than the percentage of Black individuals residing in the region (~10%) [11]. In addition, the reduction in the proportion of COVID-19 vaccination records within the region in the NYSIIS database through imputation did not significantly change the distribution of vaccinated individuals across race groups, reinforcing the persistence of this inequity in uptake within the region.

Future work for the COVID-19 vaccine campaign should further explore possible drivers of lower uptake among these populations to avoid excess burden of vaccine-preventable illness. Continued and accurate examination of the extent to which disparate uptake of the COVID-19 vaccine is prevalent is critical for evaluating the impacts of social determinants such as structural racism, and the impact that such factors have on lack of engagement with preventative care [12]. Given the continual protection of the COVID-19 vaccines against severe COVID-19 outcomes such as hospitalization and mortality, equity in vaccine uptake and engagement across all demographic groups is critical [13,14,15]. Additionally, continual efforts should be made to reduce the percentage of records with missing race values in state vaccination registries, when appropriate, to (1) avoid biased effect estimates used to inform vaccination campaigns aimed at addressing equitable uptake, and (2) to improve precision surrounding these estimates by increasing the usable sample size. Similar imputation efforts in the future should consider the prioritization of techniques assigning probabilities of race values, as compared to deterministic approaches to assigning race values, in order to respect the choices of individuals to not choose a race by which to list within their vaccination records. Finally, future efforts are warranted to identify potential reasons for missing race within vaccination records in order to limit the prevalence of missing demographic information within vaccination records, as well as to characterize those individuals most likely to not report race, to improve the accuracy of imputation work.

Continual monitoring of the usefulness and accuracy of different data solutions to address missing demographic information with various existing and reliable data sources will remain important for routine assessment of health inequities, both in the continuation of the COVID-19 vaccine campaign as well as future adult immunization efforts.

## 5. Conclusions

The absence of complete and comprehensive data detailing systematic differences in uptake of the COVID-19 vaccines by race group in the U.S inhibits the ability to ensure equity in vaccine access, confidence and receipt. The application of multiple forms of data improvement, including cross-matching and validation across a number of existing health information systems, accompanied by robust imputation techniques displays a path by which large proportions of missing data in vaccine registries can be addressed. Data improvement and imputation efforts reinforced the identification of a disparate uptake of ≥1 dose among Black individuals within the region, which highlights the possible prevalence of barriers among this population in the region. While the reduction of missing race information within vaccine records yielded similar distributions of uptake as the complete case analysis did, COVID-19 vaccine uptake is dynamic, and recurring efforts to characterize those vaccinated are important to ensure that programmatic efforts to increase vaccine uptake are targeted at the appropriate populations. Continual widespread efforts to reduce the proportion of vaccination records with missing race and ethnicity will be critical for effectively addressing barriers to equitable uptake across the U.S.

## Figures and Tables

**Figure 1 vaccines-11-00876-f001:**
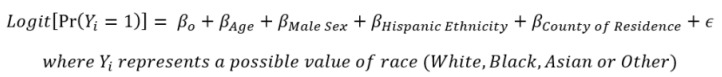
Multiple imputation multinomial logistic regression formula.

**Figure 2 vaccines-11-00876-f002:**
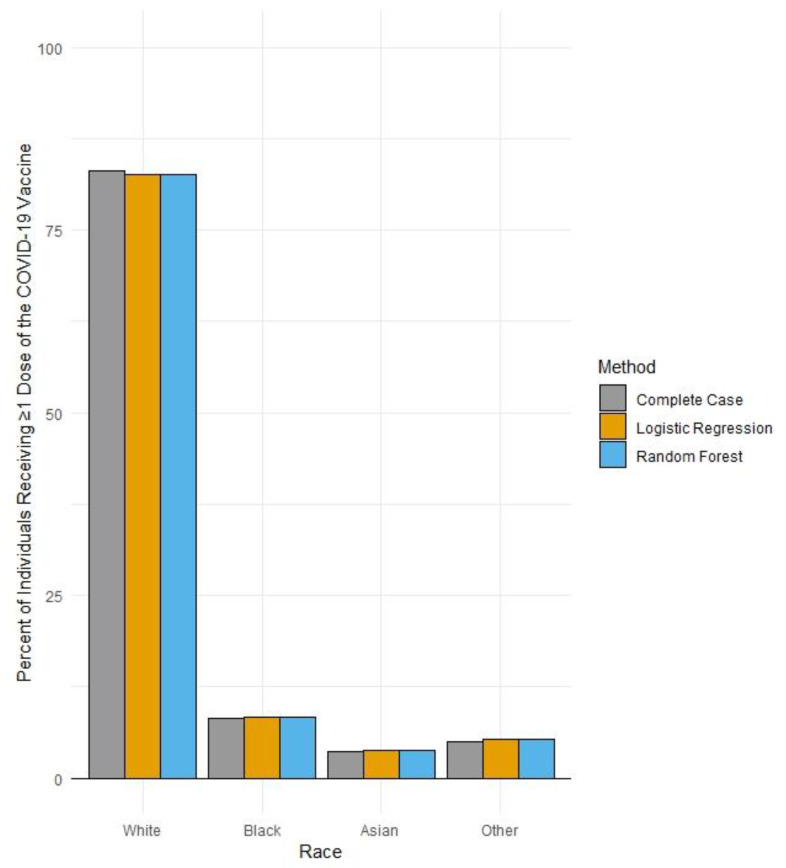
Percent uptake of >1 dose of a COVID-19 vaccine by race in the Finger Lakes region of New York by data improvement method (N = 828,551), as of 30 December 2021.

**Table 1 vaccines-11-00876-t001:** Distribution of individuals that had received ≥1 dose of the COVID-19 vaccine in the Finger Lakes region of New York by race (N = 828,551), as of 30 December 2021.

Method	White	Black	Asian	Other
	N (%)	N (%)	N (%)	N (%)
Complete Case	688,029 (83.03)	68,438 (8.26)	30,656 (3.70)	41,510 (5.01)
Logistic Regression	684,135 (82.57)	69,018 (8.33)	31,568 (3.81)	43,830 (5.29)
Random Forest	684,217 (82.58)	68,853 (8.31)	31,402 (3.79)	44,079 (5.32)

Note: all results are statistically significant due to the large sample size. However, these differences are not significant from a clinical or public health perspective.

**Table 2 vaccines-11-00876-t002:** Sensitivity analysis of distribution of individuals who received ≥1 dose of the COVID-19 vaccine in the Finger Lakes region of New York by race (N = 805,128), as of 2 December 2021, by imputation technique.

Method	White	Black	Asian	Other	Missing
	N (%)	N (%)	N (%)	N (%)	N (%)
≥1 Dose Uptake ^a^	657,870 (81.7)	71,093 (8.83)	31,158 (3.87)	45,007 (5.59)	0 (0.0)
≥1 Dose Uptake ^b^	612,088 (76.02)	63,868 (7.93)	27,572 (3.42)	39,083 (4.85)	62,517 (7.76)
Logistic Regression ^c^	663,184 (82.37)	69,241 (8.60)	29,709 (3.69)	42,268 (5.25)	725 (0.09)
Random Forest ^c^	663,828 (82.45)	68,838 (8.55)	29,548 (3.67)	42,349 (5.26)	0 (0.0)

^a^ Distribution of vaccinated individuals by race before removal of missing race values. ^b^ Distribution of vaccinated individuals by race after creating 7.76% of missing race values. ^c^ Distribution of vaccinated individuals by race with imputation of observations, in which 7.76% of race values within vaccination records were removed.

## Data Availability

Data are unavailable due to privacy limitations.

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
