# Peer review of "Bolstering the Measurement of Racial Inequity of COVID-19 Vaccine Uptake"

_vaccines, 2023, doi:10.3390/vaccines11040876_

Round 1

Reviewer 1 Report

This research article depicted a cross-sectional study mainly on the application of two multiple imputation techniques in addressing missing race values. With the calculation method described in this study, the percentage of vaccine records with missing race information was reasonably reduced to the bottom level. Was any more logical way reported to estimate the vaccine uptake distributions by race, and compared with ways in this study? In addition, how to judge if the resulting value was closest to the truth? I understand that the significance is to facilitate the vaccination campaigns, but have doubts if more significance can be summarized in the discussion, for example, any other potential application to the vaccine protection analysis. 

A minor revision in Table 1, the total percentage exceeds 100%. 

Author Response

  1. Was any more logical way reported to estimate the vaccine uptake distributions by race, and compared with ways in this study?

Response: This is a great point, we did consider a number of different denominators for the percent vaccinated, however, we arrived at the one we did (number of individuals in each racialized group for the region/total number of individuals in each racialized group for the region) so that we could directly determine if, as compared to the composition of the region for each race group included, there was a lower engagement among individuals of each group which would serve as a signal for lower engagement/possible barriers to uptake within those race groups.

  1. In addition, how to judge if the resulting value was closest to the truth? I understand that the significance is to facilitate the vaccination campaigns, but have doubts if more significance can be summarized in the discussion, for example, any other potential application to the vaccine protection analysis. 

Response: Thank you for this comment, we can discern from our sensitivity analyses described in the second to last paragraph of the methods, the last paragraph of the results, and Table 2 that our imputation techniques are quite accurate. Given this hypothesized accuracy of our findings when applied to the vaccination records, we can start to imply that our findings reinforce the presence of disparate uptake of the COVID-19 vaccine within our region among Black populations. This may signal the need for additional research to identify barriers among this population in the region in order to ensure continual engagement with COVID-19 as well as other adult vaccines. We reinforce these findings within the conclusion section of our manuscript (lines 200-206).

  1. A minor revision in Table 1, the total percentage exceeds 100%. 

Response: Thank you for catching this typo, we have fixed the error in the table and all row percentages added up to a total of 100%.

Author Response

  1. Authors should discuss the limitation of the COVID-19 Vaccine in details.

Response: This is a great point, thank you for this comment. We wanted to ensure that we pointed out what the vaccine does effectively provide protection against (COVID-19 related hospitalization and mortality) rather than focusing on lower vaccine effectiveness against symptomatic illness to avoid discouraging readers from engaging with COVID-19 vaccines. We speak to critical nature of ensuring equity of vaccine uptake to address equity of severe COVID-19 outcomes in the discussion section (lines 177-179). 

  1. Discuss the advantage part of the article in more details in the introduction section.

Response: Thank you for this suggestion, we have increased the discussion within the introduction section regarding why imputation techniques, as compared to a standard practice of complete case analysis, are advantageous for increasing efficiency of effect estimates for evaluating disparate vaccine engagement, as well as decreasing the unbiased nature of those effect estimates (lines 48-51). We also emphasize the goal of these analyses and how they will translate into practical information for targeted COVID-19 immunization campaigns within the region aimed at addressing disparate COVID-19 related morbidity and mortality by race (lines 54-57).

  1. Please polish the grammar.

Response: Thank you for this comment, we have gone through the manuscript and made the necessary changes to improve the grammar of the text.

  1. Author should provide the graphical presentation of the comparison tables.

Response: Thank you for this comment, to graphically display our primary findings in table 1, we generated a bar graph comparing the proportion of individuals within each respective racialized group who received ≥1 dose of a COVID-19 vaccine across each of the data improvement techniques discussed (complete case, logistic regression imputation, and random forest imputation).

  1. Please provide the conclusion section of the article in more details.

Response: Thank you for this suggestion, within the conclusion section of this article, we have added additional comments on 1) the important nature of the findings in terms of reinforcing the presence of disparate uptake among Black populations in the region, and 2) how these efforts for data improvement are necessary to continually engage with as the COVID-19 vaccine campaign evolves and changes over time in order to have the most accurate picture of vaccine uptake moving forward (lines 200-206).  

Reviewer 3 Report

The study applied imputation methods for the missing data on race. The results were largely the same and the actual application or value is unclear.

Comments to the authors

1.    There could be reasons of missing information on race not captured by the data or the imputation regression model. It’s not clear the imputed results were more or were less reliable.

2.    There was no validation of the results or assessment of its accuracy.

Author Response

  1. There could be reasons of missing information on race not captured by the data or the imputation regression model. It’s not clear the imputed results were more or were less reliable.

Response: Thank you for this comment, we agree it is difficult to discern what the missingness in this data indicates as well as how the accompanying results of the imputation techniques provide any additional insight into this very common phenomenon of missing race within health records. However, we believe our imputation techniques applied with the data captured in the vaccine registry used are accurate given the highly valid findings identified in the sensitivity analyses described in the second to last paragraph of the methods section, the last paragraph in the results section and in Table 2. While this doesn’t help us to understand why the data are missing, it does facilitate our understanding of where the majority of the missing race data should be categorized. We also included a discussion of how identifying the reason for missing race values would lead to improved and more accurate imputation efforts and future work to identify these reasons is warranted (lines 186-189).

  1. There was no validation of the results or assessment of its accuracy.

Response: Thank you for this comment, we did run a number of sensitivity analyses to test how accurate our imputation techniques were. This was performed by randomly removing 7.76% of vaccination records (the same percentage that were missing following cross-matching and validation across regional health information exchanges) from the complete case data. We then ran both imputation methods on these data and compared what was imputed to what was randomly removed and the accuracy was quite high with the new percentages calculated within <1% difference from complete case. These findings are described in greater detail within the manuscript in the second to last paragraph of the methods section, the last paragraph of the results section, and in Table 2.   

Round 2

Reviewer 3 Report

The authors have clarified some points. However, the study was not able to demonstrate the accuracy of imputation methods, and the study aim was not achieved.

Comments to the authors

1.    The sensitivity analysis randomly selected participants with missing race information. This is the missing complete at random (MCAR) mechanism where the results would be similar anyway with or without imputations.

2.    Hence, my previous comments remained unresolved: There could be reasons of missing information on race not captured by the data or the imputation regression model. It’s not clear the imputed results were more or were less reliable.

3.    There was no robust validation of the results or assessment of its accuracy.
